# EMS Derived Wheat Mutant BIG8-1 (*Triticum aestivum* L.)—A New Drought Tolerant Mutant Wheat Line

**DOI:** 10.3390/ijms22105314

**Published:** 2021-05-18

**Authors:** Marlon-Schylor L. le Roux, Nicolas Francois V. Burger, Maré Vlok, Karl J. Kunert, Christopher A. Cullis, Anna-Maria Botha

**Affiliations:** 1Department of Genetics, University of Stellenbosch, Stellenbosch 7601, South Africa; marlonlerous@sun.ac.za (M.-S.L.l.R.); nfvburger@sun.ac.za (N.F.V.B.); marevlok@sun.ac.za (M.V.); karl.kunert@up.ac.za (K.J.K.); 2Proteomics Unit Central Analytical Facilities, University of Stellenbosch, Stellenbosch 7601, South Africa; 3Department of Plant and Soil Sciences, Forestry and Agricultural Biotechnology Institute (FABI), University of Pretoria, Pretoria 0002, South Africa; 4Department of Biology, Case Western Reserve University, Cleveland, OH 44106, USA; christopher.cullis@case.edu

**Keywords:** water deficit stress, proteome, physiological responses, enhanced water deficit stress tolerance

## Abstract

Drought response in wheat is considered a highly complex process, since it is a multigenic trait; nevertheless, breeding programs are continuously searching for new wheat varieties with characteristics for drought tolerance. In a previous study, we demonstrated the effectiveness of a mutant known as RYNO3936 that could survive 14 days without water. In this study, we reveal another mutant known as BIG8-1 that can endure severe water deficit stress (21 days without water) with superior drought response characteristics. Phenotypically, the mutant plants had broader leaves, including a densely packed fibrous root architecture that was not visible in the WT parent plants. During mild (day 7) drought stress, the mutant could maintain its relative water content, chlorophyll content, maximum quantum yield of PSII (Fv/Fm) and stomatal conductance, with no phenotypic symptoms such as wilting or senescence despite a decrease in soil moisture content. It was only during moderate (day 14) and severe (day 21) water deficit stress that a decline in those variables was evident. Furthermore, the mutant plants also displayed a unique preservation of metabolic activity, which was confirmed by assessing the accumulation of free amino acids and increase of antioxidative enzymes (peroxidases and glutathione S-transferase). Proteome reshuffling was also observed, allowing slow degradation of essential proteins such as RuBisCO during water deficit stress. The LC-MS/MS data revealed a high abundance of proteins involved in energy and photosynthesis under well-watered conditions, particularly Serpin-Z2A and Z2B, SGT1 and Calnexin-like protein. However, after 21 days of water stress, the mutants expressed ABC transporter permeases and xylanase inhibitor protein, which are involved in the transport of amino acids and protecting cells, respectively. This study characterizes a new mutant BIG8-1 with drought-tolerant characteristics suited for breeding programs.

## 1. Introduction

Wheat (*Triticum aestivum* L.), a member of the Poaceae, has played an integral part in human civilization over the past 10,000 years due to its high nutritive value, and is still an essential cereal today [1]. It is grown in areas with erratic precipitation patterns, often leading to severe yield losses. Such losses are concerning, since the human population is increasing at an unprecedent rate, with an expected population of 10 billion by 2050 [2]. Global wheat production should, therefore, increase by approximately 60% to avoid famine [3,4]. Drought stress, due to a changing climate, causes excessive damage to wheat yield, and therefore poses an immense problem to wheat breeders.

Wheat adapts to drought conditions by shortening its lifespan. This often results in compromises at physiological and biochemical levels, leading to a change in the proteome, which ultimately affects seed quantity and quality [5,6,7,8,9,10]. Many large- and small-scale wheat farmers are suffering the consequences of prolonged drought periods. As a result, farmers require economic support [11,12,13]. Prevailing drought conditions are also often declared natural disasters that require intense management by governments to avoid significant economic and agricultural disruption. Breeding water stress tolerant wheat varieties is a priority in many arid countries [14]. Breeding for drought tolerance is, however, highly complex, owning to the fact that drought tolerance is a multigenic trait. The process is further complicated by the narrow genetic base of wheat [15,16,17,18]. Thus, several strategies, such as germplasm assessment and hybridization with wild species to obtain drought-tolerant varieties, are continuously applied to achieve this goal [19]. Ahigh degree of novel genetic variation is often achieved in crops through induced mutagenesis, either by chemicals (e.g., ethyl methanesulfonate, EMS and sodium azide) or other means (e.g., gamma radiation) [20,21], in order to improve physiological traits and increase yield, which has been reported for many crops species including wheat, barley, rice, beans, tomatoes, lentils and mango [22,23,24,25,26,27,28,29,30,31,32,33,34]. According to the IAEA database (http://mvgs.iaea.org/Search.%20aspx, accessed on 19 March 2021), two mutant wheat varieties (*Deada* and *Leana*) derived from chemical mutagenesis were successfully integrated in 2017 for direct use in commercial farming under conditions of Ukrainian Steppe farms. These varieties have improved morpho-physiological characteristics, including high protein content and drought tolerance [35,36]. Additionally, our research group developed a mutant (RYNO3936) using chemically induced mutagenesis by exposing the seed of a red hard winter wheat cultivar, Tugela DN, to an alkylating mutagen known as sodium azide. This mutant outperformed the control group but also displayed rapid recovery from water deficit conditions [37]. The ability of crops to endure various degrees of drought stress is the most sought after characteristic in many breeding programs. This fact led to the development of a nonhierarchical classification system in the early 1980s, whereby crops were classified as either avoiding, escaping or tolerating drought stress [38]. Based on this classification system, a plant’s ability to maintain a high internal water content, regardless of any depletion of water in the soil, was classified as “drought avoidance”. These plants will tightly regulate water loss and optimize their ability to take up water. They adapt their physio-morphological attributes to ensure water-use-efficiency by, for example, increasing the number of roots and adapting their root architecture. Additionally, these plants also alter their transpiration rates to avoid losing water. “Drought escape” plants within this classification system mainly rely on the ability to “sense” the drought conditions and modulate their seasonal vegetative and reproduction growth, and the plant adapts by precipitous phenological enhancement and developmental plasticity [38]. When a plant possesses an adaptive mechanism that allows changes within cells, such as cellular plasticity, osmotic adjustment or turgor pressure, these plants are classified as ‘drought tolerant’ [39].

When a plant experiences drought stress, the major components of the photosynthetic apparatus (photosystem II (PSII), the cytochrome b_6_f complex and PSI) become defective [40], which is perpetuated by chlorophyll imbalance. Chlorophyll is essential for capturing light energy, which can transfer it for the use in photochemistry or releasing the remaining energy in the form of electromagnetic radiation, often referred to as fluorescence, which is integral for the measurement of Fv/Fm [41]. The latter is, therefore, the maximum quantum yield of PSII and an indication of the function of other thylakoid membrane protein complexes, collectively forming a photosynthetic network [42]. Any compromise in photochemistry that goes beyond PSII is a causative effect of the downregulation of the photosynthetic process while stressed [43,44,45,46]. Ribulose-1,5-bisphosphate carboxylase/oxygenase (RuBisCO), a key enzyme in photosynthesis, is highly susceptible to water deficit stress. The RuBisCO protein assimilates carbon dioxide by catalysis, which results in the conversion of inorganic carbon into organic compounds. Its relative abundance is important, since the enzyme is very slow-working. Large amounts of the enzyme are, therefore, required to achieve adequate photosynthesis. A direct relationship exists between the amount of RuBisCO and the rate of photosynthesis in higher plants, such as wheat [47,48].

A further inevitable reaction is the closure of stomata during drought stress. This becomes synonymous with a decline of CO_2_ uptake, limiting the carboxylation process [49]. However, stomatal closure is regarded as a necessity to further limit internal water loss [50]. The photosynthetic carbon fixation process is further compromised by capturing more light than can be actively processed. This process prompts a surge in the production of reactive oxygen species (ROS), which disrupts photosynthesis [51]. The accumulation of ROS within cells is toxic. Such oxidative stress ultimately causes damage to proteins, DNA, lipids as well as compromising enzyme functionality and increasing membrane penetrability [52]. However, ROS production is counteracted by antioxidant systems that include a variety of antioxidative enzymes, such as superoxide dismutases, catalases, ascorbate peroxidases, peroxidases and glutathione S-transferases. The glutathione peroxidases activities of these enzymes are generally increased during drought stress. 

Stressed plants also use reconfiguration of the metabolome to provide osmo-protective functions that maintain osmotic homeostasis and prevent the degradation of enzymes [53]. In wheat, these osmolytes (amino acids, organic acids, sugars) accumulate during water deficit stress. This causes alteration of metabolic networks, for example, the citric acid cycle [53,54]. In maize, metabolic profiling revealed that a combination of carbohydrate and lipid metabolism, together with urea cycles and glutathione, are key to understanding the osmo-protective nature of the plant during drought conditions [9]. Also, when three *Triticum* species (i.e., wild emmer, *Triticum turgidum ssp. Dicoccoides;* Einkorn, *Triticum monococcum ssp. monococcum*, and hexaploid wheat, *Triticum aestivum ssp. aestivum*) were investigated, amounts of amino acids and low molecular weight compounds (glutathione) increased under drought conditions in both leaf and root tissue [55].

Drought stress further induces changes in the plant’s proteome, and several drought-stress-responsive proteins have been previously identified in wheat [56]. Many of these proteins were involved in the osmotic and ionic homeostasis, toxic by-products alleviation and growth recovery. [57] It was also found in *Triticum boeticum* that such drought-stress-responsive proteins are involved in many processes, including protein and carbon metabolism, as well as in signal transduction and photosynthesis, but also in nitrogen and amino acid metabolism. During drought, vital proteins also undergo folding associated with translation and post-translational modification, such as ubiquitination and SUMOylation [58,59]. The latter, which has been shown to have detrimental effects [60], refers to the process where small ubiquitin modifiers-1 (SUMO1) bodies bind to a spectrum of lysine residue proteins during stress and may be responsible for protein profile changes often observed between different plant systems. Modification of endogenous proteins due to the conjugation of SUMO can render them nonfunctional. SUMOylation is, however, a reversible process mediated by a cluster of SUMO proteases, also referred to as cysteine proteases (CPs) [61]. Recent studies conducted on wheat and rice successfully demonstrated the function of such CPs in both plant development and response to water deficit stress [37,61,62,63,64]. The present research aimed to study the effects of water deficit stress caused by drought conditions on an ethyl methanesulfonate (EMS)-induced mutant wheat line (BIG8-1), which has enhanced tolerance to water deficit in comparison to the near-isogenic parental line (WT) BIG8. This mutant displays unique drought tolerance characteristics, whereby it could sustain functionality on a physiological and proteomic level for 21 days without water. The overall performance of this mutant outperformed any other mutants developed in our group, including RYNO3936, which could only sustain physiological functionality for 14 days (gravimetric reading of ±24%, and RWC of ±44%). The objective of this investigation was to understand how this mutant BIG8 manages water deficit. For this purpose, the physiological and metabolic responses were characterized during the prolonged water deficit stress and a proteome analysis was conducted to follow EMS-induced changes that resulted in the enhanced water deficit stress tolerance in the mutant, not observed in its near-isogenic progenitor.

## 2. Results

### 2.1. Plant Phenotypes and Relative Moisture Content

In the first step, the phenotype of BIG8-1 mutant and nonmutated control plants were determined. The BIG8-1 mutant population (M6) displayed a stable heritability of all measurable biometric traits and had more leaves compared to the WT plants (Figure 1A,D). The mutant shoot (range: 29.00 cm–48.00 cm) and leaf lengths (range: 18.32 cm–45.00 cm), however, did not differ significantly (*p* > 0.05) from that in the WT plants (leaf lengths (range: 34.00 cm–37.00 cm) shoot length (range: 37.00 cm–26.00 cm) (Figure 2B,C) both types of plants produced similar sized seeds (Figure 1H,I). However, the leaf width of the mutant plants (range: 1.4 cm–1.7 cm) was significantly broader (*p* < 0.05) when compared with WT plants (range: 0.8 cm–1.10 cm) (Figure 2A).

When watering was withheld, the WT plants visibly wilted after four days and had senesced leaves after seven days (Day 7, Figure 1B). The plant was severely senesced and completely dead after 10 days, (Figure 1C) (gravimetric reading of ±24 and RWC of ±0.5%). In contrast, BIG8-1 mutant plants only showed visible signs of wilting after 10 days postinduction of water deficit stress (PWS), with some senesced leaves after two weeks PWS, and only died after three weeks PWS (Day 21, Figure 1G) (gravimetric reading of ±24%, and RWC of ±17.5%). Also, the mutant plants had visibly a much higher root density when compared to WT plants (Figure 1A–G).

When the RWC (relative water content) of the leaves was determined, a similar RWC for both plant types under well-watered conditions (Day 0) was found. However, with the onset of water deficit stress, WT plants had a decline in RWC, losing nearly ±60% RWC within the first seven days, and the soil moisture content dropped to 24% (Figure 3A). This was in contrast to the mutant plants, which had no significant decline in RWC during the first week PWS (Day 7), despite both having access to the same soil moisture (24%). It was only after two weeks PWS (Day 14) that a significant decline in RWC was measured in the mutant plants, which coincides with a wilted phenotype that only presents itself on day 14 (Figure 1F). During the prolonged (severe) water deficit stress (Day 21), the mutant plants only lost an additional 7% of RWC, but already displayed severe leaf senescence (Figure 1G). Despite the visible signs of advanced wilting, the stem still remained upright with no signs of senescence (Figure 1G). Similar trends were observed in the root RWC as that was observed for leaf RWC (Figure 3B).

### 2.2. Chlorophyll, Photosynthesis, Stomatal Conductance and RuBisCo Response

Next, the chloropyll concentration was investigated in the two types of plants. Under well-watered conditions, mutant and WT plants had similar total chlorophyll contents (Figure 4A). When water was withheld, a significant decline of total chlorophyll (*p* > 0.01) was found in the WT plants, but this decline was not found in the mutant plants. This decline in total chlorophyll content in the WT plants coincided with visible senescence (Figure 1B,C). In the mutant plants, significant degradation of chlorophyll only manifested after three weeks PWS (*p* < 0.05) (Figure 4A).

A consistent and steady decline in the maximum quantum yield of PSII (Fv/Fm) also occurred over the three weeks treatment period in the mutant plants. This was in sharp contrast to the WT plants, where chlorophyll fluorescence declined to almost zero within the first week PWS (Figure 4B). Stomatal conductivity measurements followed similar trends as found for chlorophyll fluorescence, as indicated by the scatter plot (Figure 4B).

RuBisCO amount also corroborated total chlorophyll content and chlorophyll fluorescence (Fv/Fm) measurements. RuBisCO quantification was conducted through protein blot analysis and probing with an anti-LSU (RuBisCO large subunit) and anti-SSU (RuBisCO small subunit) antiserum. Single cross-reacting peptides for both the large (LSU) and small (SSU) subunits were identified with sizes of 56 ± 4 kDa and 15 ± 2 kDa respectively. This corresponds to the sizes previously reported for wheat RuBisCO (Figure 5) [37,65]. 

To further quantify the relative abundance of the RuBisCO subunits, the blots were also subjected to laser densitometry (Appendix A). The densitometric data indicated that the abundance of RuBisCO LSU and SSU declined over the three weeks PWS, but not to the same extent as that observed in the WT plants, which happened within seven days.

### 2.3. SUMOylation and Protease Activity under Water Stress Conditions

To also determine post-translation modification in the plant PWS, SUMOylation using Western blot analysis was determined followed by probing with a monoclonal anti-SUMO1 antibody. A vast number of cross-reacting peptides was found, ranging in size from 80 ± 10 kDa to 10 ± 5 kDa (Figure 5). These peptides could form part of the four major classes of proteases (cysteine, serine, aspartate metalloproteases), originating from well-watered (Day 0) plant material and material from plants exposed to water deficit stress (Days 7, 14, 21). Comparative profiling of cross-reacting bands revealed that the number of SUMO1 peptides increased with increasing PWS, but this was found for both the mutant and WT plants.

Cysteine proteases (CPs) have previously been reported to reduce the detrimental effects of SUMOylation. Therefore, proteases were analyzed using zymographic assays on a one-dimensional gel. The CP inhibitor E64 was further applied to determine CPs specifically. A side-by-side zymographic comparison indicated that under well-watered conditions (Day 0), WT plants expressed more proteins with proteolytic activity than mutant plants (Figure 6A). Mutant plants further expressed three protein bands with proteolytic activity, including one protein with CP activity. After watering was stopped, more bands with proteolytic activity and particular CP activity were detected. This includes 8 proteolytic bands, including 3 bands containing CP activity detectable in Day 14 PWS and five proteolytic bands with three bands with CP activity on Day 21 PWS (Figure 6B–D).

### 2.4. Changes in the Proteome under Water Deficit Stress Conditions

In the next step, the proteome of mutant plants was compared before (Day 0) and after induction of water deficit stress (Days 7, 14, 21) (Appendix A). In the mutant plants, 64 proteins were found that were shared amongst all treatments (Figure 7), with 19 uniquely expressed on Day 0 (e.g., chloroplast 50S ribosomal protein L4, lPastocyanin, 70 kDa heat shock protein, Photosystem II reaction center Psb28 protein), 12 on Day 7 PWS (i.e., Chloroplast Ketol-acid reductoisomerase, Photosystem I P700 chlorophyll a apoprotein, Dopa-decarboxylase (Fragment), Serpin-Z2A and Z2B, Elongation factor 2, Sucrose synthase, SGT1, Calnexin-like protein, ADP-glucose brittle-1 transporter), only two on Day 14 PWS (i.e., Photosystem II 10 kDa polypeptide and the Protein IN2-1-like protein), while only three proteins were unique to Day 21 PWS (e.g., ABC transporter permease, Gamma-gliadin and Xylanase inhibitor protein I) (Figure 7).

To visualize the observed proteomic changes before (Day 0) and after induction of water deficit stress (Days 7, 14, 21), a cluster analysis [66] was conducted and the clusters were visualized using TreeView [67] (Appendix A). Two clusters were obtained with two major groupings according to protein expression patterns, with unstressed (Day 0, Cluster 2) in a separate cluster than the water deficit stressed plants (Days 7, 14 and 21, Cluster 1). Amongst the water deficit stressed plants, days 7 and 21 clustered together, while day 14 formed another grouping. The different clusters were categorized according to broad functional categories (i.e., biological processes, cellular component molecular function), and again the differences in composition between the clusters were clearly visible. For example, the control plants (Day 0) displayed fewer functional categories amongst the biological processes but more categories under the cellular component grouping, when compared with the water deficit, stressed plants (Appendix A). Also, ion binding provided the largest component of the molecular processes (Day 0), but not in the water deficit stressed plants, which contained two additional groups—enzyme regulators and structural components of ribosomes—not present in the well-watered control plants. These differences also extended to the individual subclusters amongst the different days after induction of water deficit stress.

### 2.5. Changes in Free Amino Acid under Water Stress

To also determine the effect of water deficit stress on the metabolome, free amino acids (FAA) were measured in well-watered (Day 0) and water deficit stressed (Days 7, 14, 21) leaf material (Table 1). When well-watered, the FAA levels were comparable between WT and mutant plants, except for aspartate, lysine, leucine and phenylalanine, that were higher in the BIG8-1 mutant plants. Induction of water deficit stress resulted in an overall reduction of FAAs in the mutant plants except for methionine. However, a significant increase in the amount of proline, methionine phenylalanine was measured in the WT plants. Prolonged water deficit stress increased the amount of all FAAs in the mutant plants, except for methionine where the amount decreased. The FAAs that showed the highest increase PWS whereas proline, glutamine and aspartate.

### 2.6. Oxidative Defense 

When the activity of antioxidative enzymes was measured, the activity of GST was comparable between the WT and mutant plants under well-watered conditions (*p* > 0.05). However, water deficit stress significantly induced GST activity in the WT plants, but not in the mutant plants (Figure 8A). GST activity further increased significantly in the mutant after prolonged water deficit stress (Day 14), whereafter activity decreased (Day 21). For POX activity, there was a significant difference between the WT and mutant plants under well-watered conditions (*p* < 0.001). A substantial decrease in POX activity was, however, evident seven days PWS in the WT plants (Figure 8B). POX activity increased in the mutant with prolonged water deficit stress (Day 14) but decreased under severe water deficit stress conditions (Day 21).

## 3. Discussion

In the current study, we characterized in greater detail a new chemically-induced wheat mutant line, BIG8-1, showing favorable drought tolerance characteristics. We specifically determined the physiological and biochemical responses of mutant plants under well-watered, mild and severe water deficit stress conditions. This study extended our previous study already characterizing the drought-tolerant wheat mutant line RYNO3936. This new mutant importantly presented a different phenotype as it could only endure drought stress for longer. The RYNO3936 mutant and BIG8 stem from the similar genetic backgrounds. However, the fundamental difference is that RYNO3936 was developed using 1 mm sodium azide, whereas BIG8-1 was developed by treatment with 1 mM EMS. The findings in this study revealed that BIG8-1 mutant plants could endure severe water deficit stress (21 days without water) using some of the classical strategies described as “drought avoidance” [38]. This refers to the mutant’s ability to maintain a high tissue water content to ensure metabolic activity during water deficit conditions, which was possible due to a mutated proteome. Drought avoidance, by definition, allows plants to maintain higher internal water content despite the depletion of water in the soil [68]. These mutants changed phenotypically, and adjusted their physiological and biochemical strategies to alleviate stress by, amongst others, changing its proteome. These changes further allows for slow degradation of essential proteins during the varying degree of induced water deficit stress. 

Under well-watered conditions, these plants did not differ much in terms of their above-ground features, except for the mutant plants having more leaves, which were also broader—an attribute that is associated with drought adaption, for enhancement of photosynthesis, internal water retainment and gas exchange [69]. However, the plants had different root architectures, with the mutant plants possessing a densely packed fibrous root system (Figure 1D) which was not visible in the WT parent plants. Developing more root biomass is a typical response to drought stress in plants, allowing them to take up more water from the soil. The visibly larger root mass presumably creates a larger absorptive surface area per gram dry weight of root [70,71]; however, further investigation is needed in this regard. Overall, these morphological changes may be a consequence of random mutagenesis, rather than of the hexaploidy ancestral heritage of wheat.

The first phenotypic change that wheat plants suffer when exposed to water deficit stress conditions is wilting, which is indicative of internal water potential imbalance due to the lack of soil water [72]. This is a necessary trait to prevent further water loss by changing the leaf morphology (rolling) to reduce the area of transpiration, thereby preserving water [69]. The mutant plants displayed no phenotypic symptoms such as wilting or senescence when exposed to mild water deficit stress (Day 7). This is mainly due to the ability to maintain the same RWC under being well-watered (±70%) to being exposed to mild water deficit stress (±68%) despite the drastic change in soil moisture content (±85% to ±21%) (Figure 3). The mutant most likely has the ability to increase its water-use-efficiency, due to an elaborated rooting system; therefore, the mutant could maintain RWC, even under mild drought conditions. It was only during advanced stages of the moderate water deficit stress (Day 14), when the RWC was ±31%, that wilting became visible in the mutant plants, as they suffered leaf drooping with slight rolling (Figure 1F). Prolonged, severe water deficit stress (Day 21) also induced leaf folding and rolling with leaf senescence and loss of pigmentation (Figure 1G). However, the stem of mutant plants seemed to remain alive, which was distinguishably by pigmentation and elasticity. This unique and selective wilting was not observed in WT plants, which suffered complete wilting by day 7 (leaves and stem). This also suggests that the mutant plants may have a slow wilting phenotype. This phenotype can be attributed to the ability to conserve water from the soil under well-watered conditions [73] which is drawn upon under water-limiting conditions (Figure 3). This corroborates our finding, as the mutant plants differ significantly from the WT plants with regard to RWC under well-watered conditions, which then is slowly depleted with the prolonged periods of water stress.

Chlorophyll a/b content was reported to decrease in water-stressed wheat [74,75,76]; this parameter can be a valuable indicator for assessing how the plant manages environmental stress, and also serves as a determining factor for biomass accumulation. Total chlorophyll in the mutant did not show significant changes with the induction of water deficit stress in the mutant plants when compared with the WT plants (Figure 4). Chlorophyll content only significantly declined with prolonged water-stress when most leaves changed from green to yellow (Figure 1F,G). These symptoms are likely due to increased peroxidation of chloroplast membranes and electrolytic leakage from the thylakoid membrane [77]. The observed reduction in chlorophyll pigment was also previously reported and is associated with the combined effect of chlorophyll loss and water stress, especially in drought-sensitive wheat varieties. In contrast, better preservation of chlorophyll was found in drought-tolerant plants, emphasizing their capacity to have optimal photosynthesis despite water stress conditions [23,78,79]. 

Photosynthetic pigments are instrumental in allowing the plant to absorb energy from sunlight, and foliar chlorophyll content is a determining factor in photosynthetic rates [80]. Although we did not demonstrate a statistically significant relationship between chlorophyll content and the rate of PSII efficiency (Fv/Fm). Our data does show a concomitant change of these two physiological parameters over time. Therefore, based on the chlorophyll content measured in the mutant plants relative to WT plants, a high rate of PSII efficiency (Fv/Fm) was observed (Figure 4). Such change in Fv/Fm when plants are exposed to stressful environments is well-documented in various crops, such as maize [81], rice [82], beans [83] and tomatoes [84]. It has been concluded that the PSII reaction center may have suffered some sort of impairment by photoinactivation that is associated with oxidative damage [42,70,85]. Nevertheless, the mutant maintains an unusually high rate of photosynthesis during the onset of the water deficit stress period (Days 7 and 14) when compared with the WT, which may suggest that either structural modification protects components of the photosynthetic apparatus due to mutations, or alternatively, there is the active involvement of protective enzymes in the process.

Plant adaptability to drought stress is also reliant on the ability to enhance its antioxidative mechanism to avoid the accumulation of reactive oxidative species (ROS). Production of such species can often lead to the initiation of uncontrollable oxidative cascades that damage cell membranes and other cellular components resulting in eventual cell death [77,86]. Plants have developed strategies to minimize the deleterious effects of ROS, amongst which ROS-scavenging enzymes such as POX and GST are directly involved in the antioxidative-stress response [87,88]. The mutant BIG8-1 seemingly does not respond metabolically to mild levels of induced water deficit stress, as no significant changes in ether POX or GST activity levels was found (Figure 8). However, changes were evident in the WT plants (*p* < 0.05.). With the prolonged water deficit stress (Days 14 and 21), ROS accumulation likely reached cellular toxicity leading to an increase in POX and GST activities. Where POX catalyzes hydrogen peroxide for the oxidation of a wide range of substrates that tend to increase with water deficit stress, mainly phenol derivatives [89,90] and GST provides unique intracellular protection with sustaining cell redox homeostasis [20,88]. To further explore if the photosynthetic activity was changed in mutant plants, RuBisCO abundance was measured. RuBisCO consists of eight large and eight small protein subunits. The chloroplastic genome contains the *rbcL* gene that encodes the large subunit (LSU). A family of *rbcS* genes is nestled in the nuclear genome, which encodes several variants of the small subunit (SSU) [91]. Under mild water deficit stress, both RuBisCO subunits were affected. (Figure 5). A decrease in the relative abundance of both LSU and SSU was, however, only measured after prolonged water deficit stress, while WT plants had a rapid degradation of both LSU and SSU (Days 7 and 14). In general, RuBisCO is mobilized through proteolysis during senescence or oxidative stress to contribute to the intracellular reservoir of nitrogen [92,93]. The mutant plants showed a slower and more gradual decline in RuBisCO abundance (Figure 5, Appendix A), which demonstrates that the BIG8-1 plants were able to configurate their degradome to ensure that important proteins, such as RuBisCO, remain available and also to ensure sufficient nitrogen supply [92]. Furthermore, a concomitant reduction of both LSU and SSU was found in mutant plants. Both subunits are generally present in equal quantities to match the one-to-one RuBisCO large and small subunit stoichiometry [94]. In a previous study, the preferential degradation of the LSU in another drought-tolerant mutant line, RYNO3936, was found. These RYNO3936 mutants also expressed a high recovery capacity after senescence [37].

Protein abundance is grossly affected by drought stress and mediated by various proteases. There are four major classes of proteases: serine, cysteine, aspartate and metalloproteases [95]. CPs are well-characterized when compared with the other proteases. However, wheat CPs and their expression still require further study, especially in the context of drought [96]. Only prolonged water deficit stress in mutant plants lead to an increase in the number of CPs expressed. This also provides an indication that these mutant plants are more drought-tolerant and can better limit any degradative proteolytic events. An increase in the number of proteases signifies an active defense response and their involvement in the degradation of storage proteins for the release of amino acids, which then serve as substituents in the biosynthesis of new proteins [97]. However, proteases can also be involved in the protection of crucial proteins by facilitating post-translational modification, such as in the case of SUMOylation [98].

SUMO modification intrinsically modulates intracellular localization, protein-protein relationship, or the activities of a protein during both stressed and nonstressed conditions [99]. The SUMO attachment to substrates is a sequential enzyme reaction that involves the SUMO activating enzyme (E1), SUMO conjugating enzyme (E2) SUMO ligase (E3) [100]. In our study, focus was on the hyperconjugation of nonspecific proteins during prolonged water deficit stress, which has been shown to have detrimental effects on plant development [98,101]. However, CPs can circumvent the effects of SUMOylation [65,96]. Mutant plants showed a gradual increase in SUMOylation across the different time points PWS (Figure 5), an expected consequence since there was effective management of proteases as illustrated with zymography (Figure 6). 

A significant change in the accumulation of free amino acids (FAA) was also found. Free intracellular amino acids are the “currency” through which protein metabolism operates. Their intracellular distribution is one of the significant factors in the regulation of protein metabolism. With the induction of water deficit stress, mutant plants decrease their FAA content (apart from methionine that increases) (Table 1). Exposure to prolonged water deficit stress (Days 14 and 21), greatly increased the amount of all FAAs with the exception of methionine. However, substantial increases were seen for aspartate and glutamine, with the most significant change for proline. An increase in protein degradation normally leads to an increase of free methionine [102,103]. Methionine is, however, highly sensitive to oxidative processes leading to the formation of methionine sulfoxide [104]. This formation has a detrimental effect on the biosynthesis of different proteins affecting physio-biochemical attributes [104,105]. Our results provide evidence that such drought-dependent oxidative processes were likely reduced in the mutant plants, which ultimately resulted in overall better drought tolerance of these plants.

Reprogramming of the proteome is also an important coping mechanism to alleviate stress. In the proteome analysis of mutant plants before and after prolonged water deficit stress, a high abundance of proteins was found which is involved in energy and photosynthesis under well-watered conditions (Appendix A). Particularly Serpin-Z2A and Z2B, SGT1 Calnexin-like protein. These proteins are known for their function in signal transduction and/or stress modulation. For example, the serpin family functions through irreversible inhibition of endogenous and exogenous proteinases, which play important roles in plant growth, development, stress responses defense against insects and pathogens [106]. Serpins are also significantly expressed under osmotic conditions such as salt and cold stresses [4]. SGT1 is known for its involvement in host defense [107], while Calnexin-like protein is a binding protein that triggers signal transduction processes that modulate cellular function [108]. ABC transporter permeases were expressed on Day 21, where they are likely responsible for transporting amino acids to various parts of the plant and may be important for nitrogen cycling [109]. A Xylanase inhibitor protein I was also expressed during the prolonged water deficit stress (Day 21). Plant cells are protected from their surrounding environment by the cell wall, forming a structurally heterogeneous barrier. But the cell wall can be degraded by xylanases (also referred to as endo-β-1,4-xylanases or endoxylanases, EC 3.2.1.8), that depolymerize xylan, which, next to cellulose, is one of the most abundant polysaccharides in the cell wall of higher plants. Producing a xylanase inhibitor protein I may prevent the degradation of xylan and assist in stabilizing the structure of the cell wall [110].

Collectively, it was found that BIG8-1 mutant plants differed significantly from near-isogenic parent WT BIG8 plants, including the previously characterized RYNO3936 mutant regarding tolerance to water deficit stress (Appendix A). The mutant plants remained viable for up to 21 days PWS, in contrast to WT plants that died after 10 days of water deficit stress. The reasons for better drought tolerance include the facts that the mutant plants maintained leaf and root RWC at much higher levels for a longer period when compared to WT plants, despite low soil moisture availability. This is very likely due to the denser root mass, but also proteome reprogramming, since unique proteins were expressed within BIG8-1, which likely contributed to altered metabolism. However, the extent to which the proteome was mutated has not yet been investigated. Nevertheless, the mutant had enhanced physiological traits, such as in the case of maintaining chlorophyll content and PSII efficiency for longer, in addition to a delay in visible senescence. Delayed senescence was also evident from delayed protein degradation in the mutant plants when compared with WT plants. Finally, CPs very likely had an important function in sustaining functionality and regulating protein degradation to stabilize protein networks in the mutant plants. As the mutant BIG8-1 showed sustained metabolic activity and delayed senescence for two weeks, PWS was not observed in the unmodified WT BIG8 line that died within 10 days PWS. This new mutant now adds to a start-up mutant library, wherein it represents the line with the best drought tolerant traits, making it ideal for dryland conditions.

## 4. Material and Methods

### 4.1. Plant Materials, Growth Conditions Water Deficit Stress Treatments

Random mutagenesis was performed as previously described [111]. In brief, mutant BIG8-1 was developed using EMS treatment (1 mM EMS for 2 h), whereafter the treated seeds were planted in trays containing equal amounts of the substrate (sand: soil) and grown in a greenhouse at temperatures between 20 °C to 26 °C. After a month of growth, water was withheld, and plants were selected for water deficit tolerance under low nitrogen regimes. The resultant mutant, BIG8-1 shown signs of drought tolerance and outperformed the control with all physiological assessments was selfed for six generations with selection to retain the drought-tolerance trait [111].

Seeds of wildtype (WT) BIG8 and EMS mutant BIG8-1 were planted in a greenhouse with natural day/night temperature at 23 ± 3 °C (Welgevallen Experimental Farm, Stellenbosch University, South Africa). Thirty pots [dimensions: 25 cm (diameter × 30 cm (height)] containing equal amounts of sand and crusher dust (1:1) were planted with five seeds in each pot. The pots were arranged in a randomized complete block design. To water the plants, a fully automated system containing nutrients (Multifeed ^TM^. South Africa) was applied. Similar plant height (90–100 cm) and growth stages were assured between the mutant and the WT plants, which corresponded to the Zadoks’ scale stages [112]. The pots were continuously assessed by measuring soil moisture content to confirm a constant gravimetric reading of 80% until the plants reached final extension (58–65 days after germination) corresponding to phase 45 of 196 Zadoks’ scale [112,113]. At this stage measurements were collected (Day 0), water was then withheld for a total of 21 consecutive days with measurements taken on a weekly basis (Days 7, 14, 21) or until soil moisture reached 21–24%.

### 4.2. Measurement of Plant Growth and Relative Water Content

The leaf (length, width) and shoot (height) dimensions were determined with a measuring tape (unit of measurement in cm). Plant height was measured from the tip of the tiller to the ground (*n* = 20). Triplicate samples were prepared for the measurement of relative water content (RWC). Leaf and root material were assessed at each of the respective time points (Days 0, 7, 14, 21) using three similar-sized leaves and six replicates. The third leaf was cut to determine the fresh weight (FW), and then placed in deionized water for 24 h at room temperature under low-light conditions. After soaking, leaves were blotted dry with tissue paper, and turgidity was measured (TW). The same leaf was subjected to complete dehydration by placing it in a benchtop oven at 80 °C for 16 h to quantify the dry weight (DW). RWC was calculated using the following equation: RWC (%) = (FW − DW)/(TW − DW) × 100% [114,115]. Soil samples (*n* = 3) were collected at a depth of 150 mm and dried in an oven at 105 °C for 48 h after which the soil was again weighed and the gravimetric soil moisture content determined [116]. Wilting of both the mutant and the WT plants was assessed at each of the time points following the categories described [117].

### 4.3. Chlorophyll Fluorescence (Fv/Fm)Stomatal Conductance Chlorophyll Content

Stomatal conductance and chlorophyll fluorescence were recorded at each of the time points (Days 0, 7, 14, 21) as previously described [65]. Stomatal conductance was measured at three locations (*n* = 3) on the same leaf (*n* = 3) using three independent plants (*n* = 3) and a porometer (model SC-1, Decagon Devices Inc., Pullman, WA, USA) following the manufacturer’s instructions. Chlorophyll fluorescence was measured using a hand-held Chlorophyll Fluorometer (model: OS-30P; Manufacturer: Opti-Sciences, Inc, Hudson, NY United States) following the protocol by [118] and using a leaf that was fixed with dark adaptation clips for 20 min prior to reading. The latter was a prerequisite in order to achieve a flush out of assimilates. Both instruments were applied at different locations on the same flag leaf (tip to the base) to represent the entire leaf surface. Biological (*n* = 3) and technical (*n* = 3) repeats were conducted at the respective time points (Days 0, 7, 14, 21). Chlorophyll concentrations were quantified (*n* = 3) and calculated according to [119] using the SmartSpec™ Plus BioRad (Sigma-Aldrich, St. Louis, MO, USA).

### 4.4. SDS-PAGE Electrophoresis and Western Blot Analysis

Total protein was extracted (*n* = 3) and separated using the Mini-Protein TGX gradient gel (4–15%), as previously described [65]. The Bio-Rad protein assay reagents with bovine albumin as the standard (Bio-Rad Laboratories Inc., Hercules, CA, USA) were used for the determination of protein concentration [120], and quantified using a Glomax Spectrophotometer (Promega, Sunnyvale, CA, United States) [121].

Western blot analyses were conducted using a Bio-Rad Trans-Blot^®^ SD semi-dry transfer cell apparatus and polyvinylidene difluoride membranes (Hybond-P; Amersham Biosciences Ltd, Buckinghamshire, UK.) [37]. The membranes were blocked with 3% bovine serum albumin (BSA) and probed with polyclonal large (RbcL) and small (RbcS) RuBisCO Subunit IgG (RbcL and RbcS, 1:7000 [112]) and human anti-SUMO1 monoclonal antibody (1:10,000) (UBPBio, Aurora, CO, USA) diluted in buffered saline containing 3% BSA. Detection employed alkaline phosphatase-conjugated Donkey Anti-Mouse (Abcam) (1:2500) or goat anti-rabbit (1:7000) (Sigma-Aldrich, St. Louis, MO, USA) antibodies in conjunction with nitro blue tetrazolium and 5-Bromo-4-chloro-3-indolyl phosphate (Sigma-Aldrich, St. Louis, MO, USA).

### 4.5. Cysteine Proteases Activity

Total proteases were extracted from leaf material (*n* = 3) that was flash frozen in liquid nitrogen. Cold 0.1 M citrate-phosphate buffer (pH 5.6) containing 10 mM L-cysteine was added to the powder and 5× *g* at 25,000× *g* for 20 min at 4 °C. Electrophoresis was applied to the supernatant using a gradient acrylamide gel (5–15%) created by Hoefer™ SG Series Gradient Makers and Bio-Rad casting [65]. A pre-electrophoresis step was included to serve as a gel equilibration step, at 50 V for 60 min in the gel buffer storage condition at 4 °C. Cysteine protease inhibitor E64 was loaded with or without samples (80 mg) [122,123], and for 2 h, proteins were electrophoresed at 15 mA. Following electrophoresis, the glass plates were separated to free the gels and then washed in a renatured buffer (5 mM cysteine and 2.5% *v/v* Triton-X 100 ) and subsequently incubated in developing buffer (0.5% *v/v* Triton-X 100, 50 mM Tris–HCl, pH 7.5 and 5 mM CaCl2, 1 mM ZnCl2, 10 mM cysteine) for 24 h. The gels were stained with Coomassie R-250 and de-stained until clear zones were visible against the dark blue background [65]. Gels were digitalized using Gel Doc XR+ System and imported to Microsoft PowerPoint 2016 (KB4011564) 64-Bit Edition, where the entire gel image was adjusted to +40% and brightness to +20% and finally cropped for presentation purposes.

### 4.6. Amino Acid Extraction and Quantification

Leaf material was collected (*n* = 3) and desiccated in a benchtop oven at 60 °C for 24 h. Then, the samples were finely grounded, and 0.5 mL of 6 M HCI containing norleucine (250 ppm) was added, the latter serving as an internal standard. The extraction of amino acids was derivatized using the AccQ.Tag Ultra Derivatization Kit following the manufacturer’s instruction (Waters), and separated as described [37]. The concentration of amino acids in each sample was calculated based on the peak areas and calibration curves generated with commercial standards.

### 4.7. Protein Extraction, Quantification Digestion

All proteome analyses were conducted using three biological repeats (*n* = 3). Plants were exposed to well-watered (Day 0) and water deficit stress conditions (Days 7, 14, 21) after which leaf protein (*n* = 3) was extracted using a modified method [37]. After extraction, the pellet was lyophilized for two hours and stored at −80 °C until further use. The iTRAQ labelling and SCX fractionation was conducted as previously described [37] by the Proteomics Unit, Central Analytical Services, Stellenbosch University. Liquid chromatography was performed on a Thermo Scientific Ultimate 3000 RSLC equipped with a 0.5 cm × 300 µm C_18_ trap column and a 35 cm × 75 µm in-house manufactured C_18_ column (Luna C_18_, 3.6 µm; Phenomenex) analytical column. The Thermo Scientific Fusion mass spectrometer was equipped with a Nanospray Flex ionization source. Data were collected in positive mode with spray voltage set to 2 kV and ion transfer capillary set to 275 °C. Spectra were internally calibrated using polysiloxane ions at m/z = 445.12003 and 371.10024. MS1 scans were performed using the orbitrap detector set at 120,000 resolution over the scan range 350–1650 with AGC target at 3 E5 and maximum injection time of 40 ms. Data were acquired in profile mode. MS2 acquisitions were performed using monoisotopic precursor selection for ion with charges +2 − +6 with error tolerance set to ±0.02 ppm. Precursor ions were excluded from fragmentation once for a period of 30 s. Precursor ions were selected for fragmentation in HCD mode using the quadrupole mass analyzer with HCD energy set to 32.5%. Fragment ions were detected in the orbitrap mass analyzer set to 15,000 resolution. The AGC target was set to 1E4 and the maximum injection time to 45 ms. The data were acquired in centroid mode. The raw files generated by the mass spectrometer were imported into Proteome Discoverer v1.4 (Thermo Scientific) and processed using the SequestHT algorithm included in Proteome Discoverer. Data analysis was structured to allow for methylthio as fixed modification as well as NQ deamidation (NQ), oxidation (M). Precursor tolerance was set to 10 ppm and fragment ion tolerance to 0.02 Da. The database used was the murine taxonomy data base obtained from Uniprot with the sequence of amyloid beta A4 P05067 added. The results files were imported into Scaffold v1.4.4 and identified peptides validated using the X!Tandem search algorithm included in Scaffold. Peptide and Protein validation were performed using the Peptide and Protein Prophet algorithms. Protein quantitation were performed by first performing a t-test on the paired data and applying the Hochberg-Benjamini correction [124]. 

Differential expression of peptides between treatments (i.e., Days 0, 7, 14 21) was analyzed using Scaffold Viewer 4 proteomics software (http://www.proteomesoftware.com/products/scaffold/;Searle, accessed on 19 March 2021) by comparing all treatments with each other. The Benjamini-Hochberg multiple testing adjustment was applied in order to control the comparison-wide false discovery rate [124]. Sequences representing the peptides were subjected to Blast2GO [125] analysis to obtain the representing genes, as well as gene ontologies and functional categories. A *p*-value ≤ 0.05 was used as the threshold to determine the significant enrichments of GO and KEGG pathways. To cluster the data, peptide intensity signals were first normalized using the Cluster program [66], with mean-centering applying Spearman’s rank correlation. A cluster image representing groups of differentially expressed peptides that share similar expression patterns were generated from the normalized data and visualize with Java TreeView [67]. 

### 4.8. Enzyme Measurements

The extraction of enzymes was performed as described in [126]. All enzyme activity measurements were conducted using three biological repeats (*n* = 3) and in triplicate (*n* = 9). Peroxidase (POX) activity was determined following a modified method of [127] with 0.1 M sodium phosphate buffer (pH 5), 3 mM H_2_O_2_, 3 mM guaiacol an aliquot of the enzyme extract [128]. The formation of tetraguaiacol was monitored at 470 nm. POX activity was expressed as mmol tetraguaiacol min^−1^ mg^−1^ protein. Glutathione S-transferase (GST) enzyme activity was measured as described by [129], using 0.1 M phosphate buffer (pH 6.5), 3.6 mM reduced glutathione, 1 mM 1-chloro-2,4-dinitrobenzene (DNB) an aliquot of the enzyme extract [128]. The formation of GS-DNB conjugate was monitored at 340 nm. GST activity was expressed as mmol GSH min^−1^ mg^−1^ protein. 

### 4.9. Statistical Analyses

All data were collected using three biological repeats (*n* = 3) with measurements done in triplicate (*n* = 9). Mean values are presented with their standard deviation (SD) and analyzed using Graphpad Prism software version 5.0 [130]. Statistical validation and significance (*p*  ≤ 0.05) were determined with a one-way analysis of variance followed by post- Dunnett’s or Turkey or Bonferonni Test (http://www.graphpad.com/scientific-software/prism/, accessed on 19 March 2021).

## Figures and Tables

**Figure 1 ijms-22-05314-f001:**
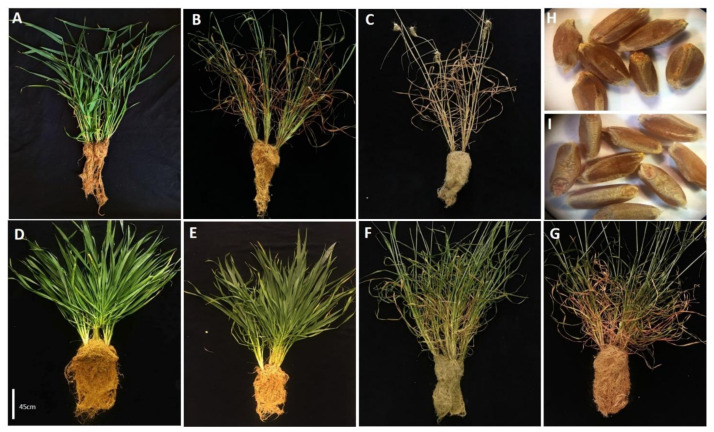
The phenotype of the wildtype (WT) BIG8 and Mutant BIG8-1 plants under well-watered (**A**,**D**) and water deficit stressed (**B**,**C**,**E**–**G**) conditions. WT plants grown under well-watered (Day 0) (**A**) and exposed to water deficit stress conditions for 7 (**B**) and 14 (**C**) days, however the WT plant was dead by Day 10. Mutant plant grown under well-watered (Day 0) (**D**) and exposed to water deficit stress conditions for prolonged periods (Day 7), (**E**), Day 14 (**F**) and Day 21 (**G**). Also shown are seed produced by the WT (**H**) and mutant (**I**) plants. Scale bar = 45 cm.

**Figure 2 ijms-22-05314-f002:**
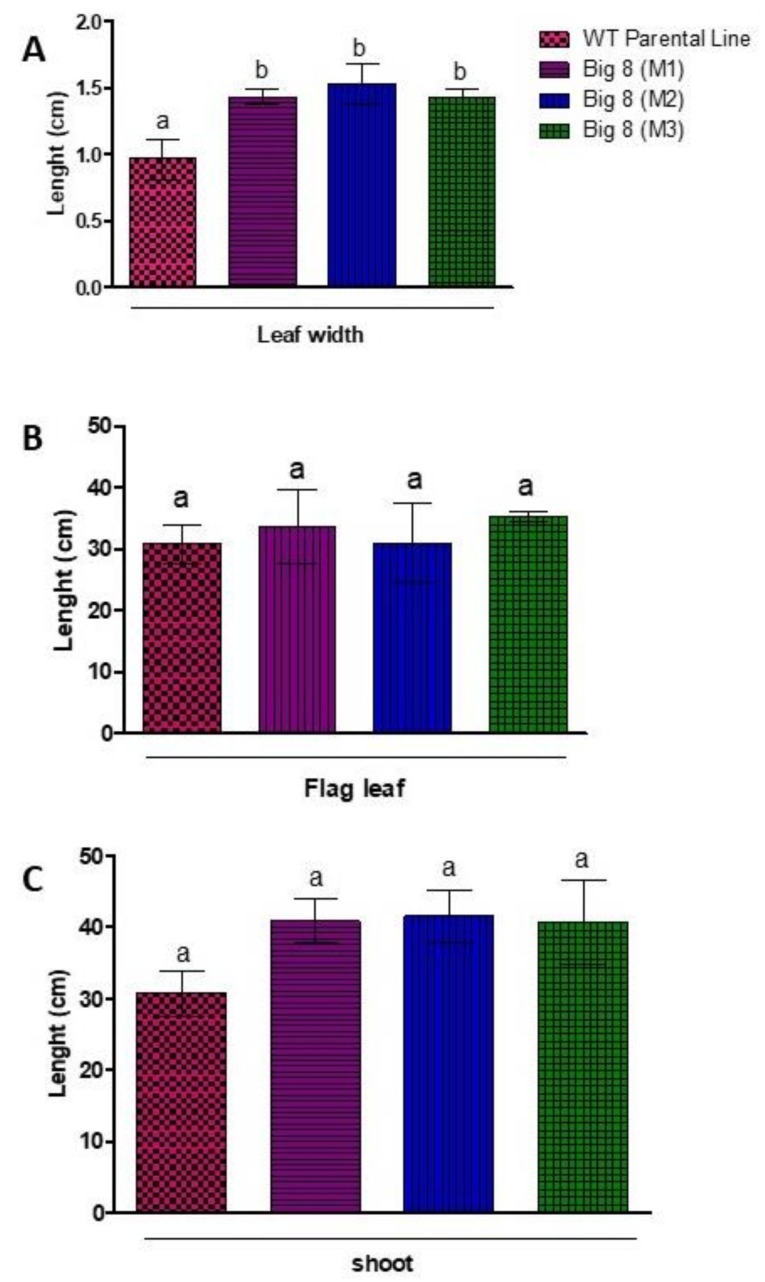
Physical measurements of flag leaf width (**A**) and length (**B**) and plant height (**C**) of WT plants compared to BIG8 mutant plants under well-watered conditions (Day 0). Results are the mean ± SD of three replicates. Similar letters on bars indicate no significant difference and different letters indicate significance at *p* ≤ 0.05.

**Figure 3 ijms-22-05314-f003:**
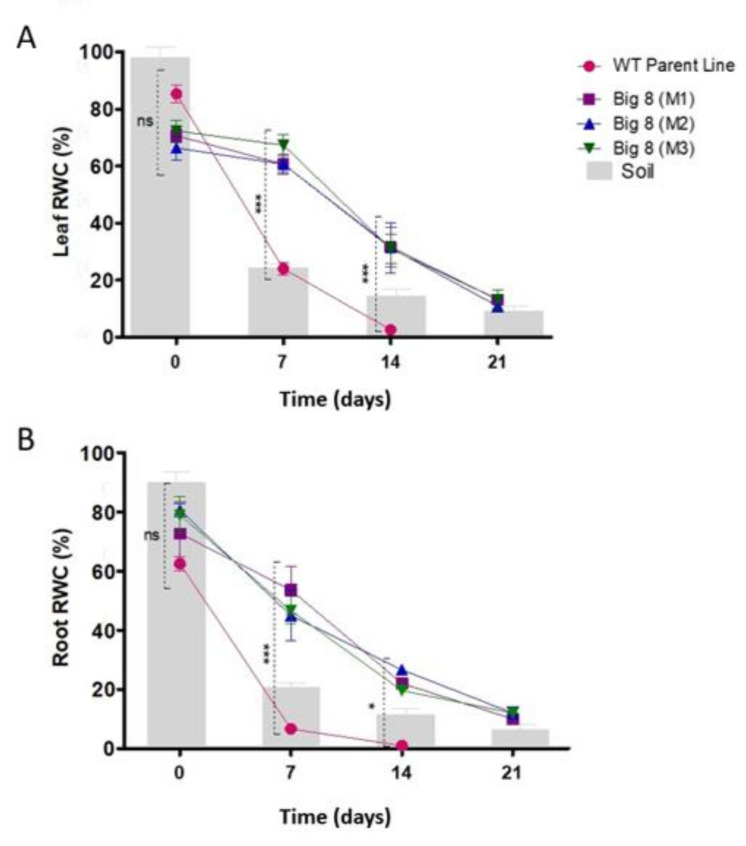
Comparison between relative water content (RWC) of leaves (**A**) and shoots (**B**) of plants under well-watered (Day 0) and exposed to water deficit stress conditions (Day 7, 14, 21 PWS). RWC is superimposed over the gravimetric readings of soil. Results are the mean ± SD (*n* = 9) and asterisks indicate a statistically significant difference between WT parental line and mutant BIG8 in the same conditions * *p* ≤ 0.05, *** *p* ≤ 0.001. Although the WT parental line was dead at 10 days, the data were collected at day 14 and compared against mutant for statistical representation.

**Figure 4 ijms-22-05314-f004:**
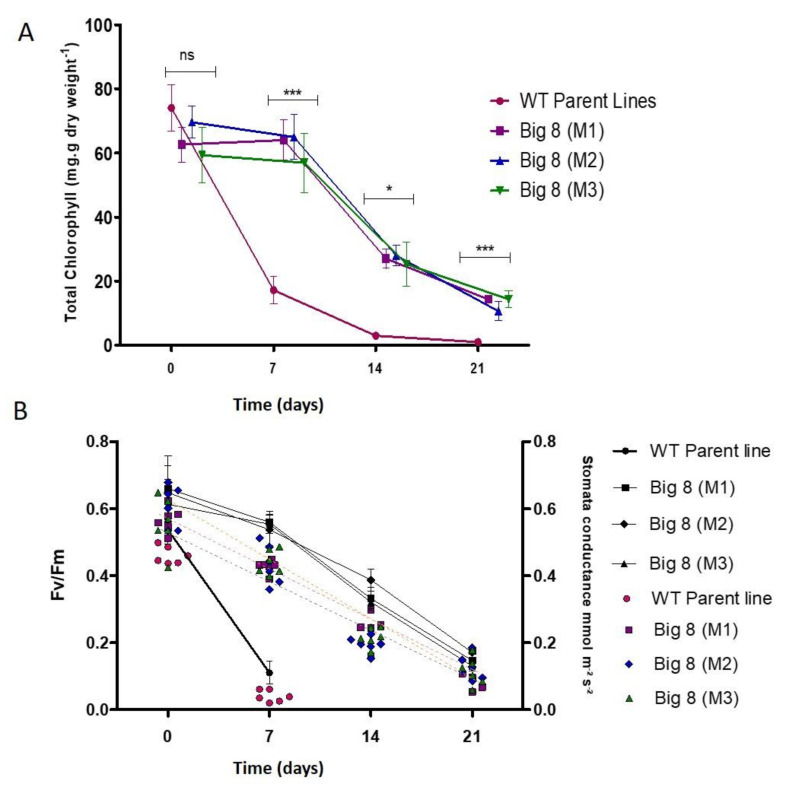
(**A**) Changes in total chlorophyll content in WT BIG8 and mutant BIG8-1 plants under well-watered (Day 0) and after exposure to water deficit stress conditions (Day 7, 14, 21 PWS). Each bar represents the mean ± SD of three replications. Asterisks indicate statistically significant differences between WT BIG8 and mutant BIG8-1 plants under similar conditions * *p* ≤ 0.05. *** *p* ≤ 0.01 and *ns* refers to non-significance(**B**) Changes in the rate of PSII efficiency (Fv/Fm) (line graph), and stomatal conductivity (scatter plot) in the WT BIG8 and mutant BIG8-1 plants under well-watered (Day 0) and after exposure to water deficit stress conditions (Day 7, 14, 21 PWS).

**Figure 5 ijms-22-05314-f005:**
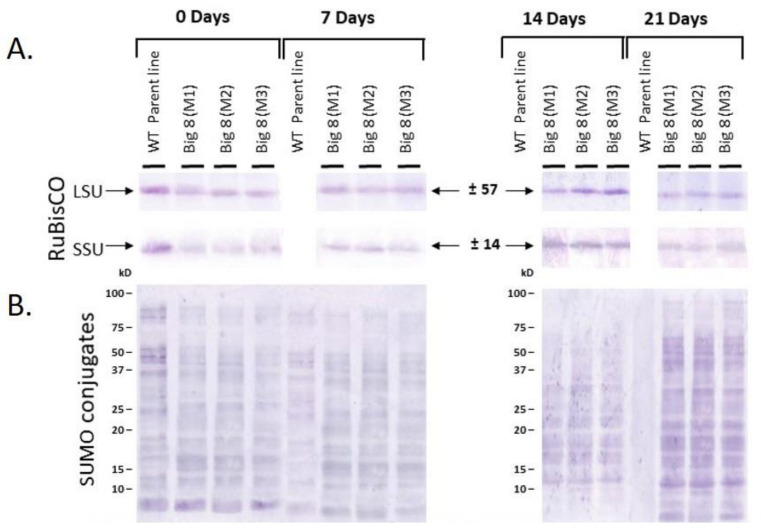
(**A**) Western blot analyses of Rubisco large (LSU) and small (SSU) subunit, as well as (**B**) SUMO conjugates extracted from leaves of WT BIG8 and mutant BIG8-1 plants under well-watered (Day 0) and after exposure to water deficit stress (Day 7, 14, 21 PWS). Each lane contains 20 μg protein. The proteins were transferred to a nitrocellulose membrane and probed with polyclonal IgG raised against LSU and SSU (1:7000 dilution); while in the case of the SUMO conjugates with a monoclonal IgG raised against SUMO1 (1:10,000 dilution). M1–M3 represents result from three independent mutant plants.

**Figure 6 ijms-22-05314-f006:**
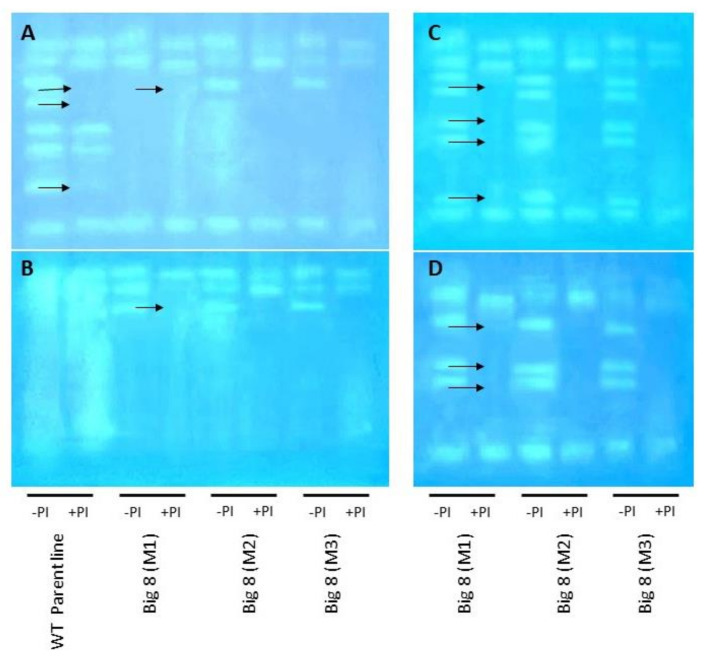
Proteolytic activity of WT BIG8 and mutant BIG8-1 plants under well-watered (Day 0) and after exposure to water deficit stress. Gradient zymograms (5–15%) were loaded with 35 μg protein per lane. An incubation step with 0.1 mM cysteine protease inhibitor (E-64) was included to allow for the identification of cysteine proteases (CPs) during (**A**) Day 0, (**B**) Day 7, (**C**) Day 14 and (**D**) Day 21. + PI indicates the lanes with the protein that was pre-treated with 0.1 mM cysteine protease inhibitor (E-64), − PI indicates no incubation with E-64. Arrows indicate absence of bands due to treatment with the 0.1 mM E-64. Presented data are representative for two independent experiments.

**Figure 7 ijms-22-05314-f007:**
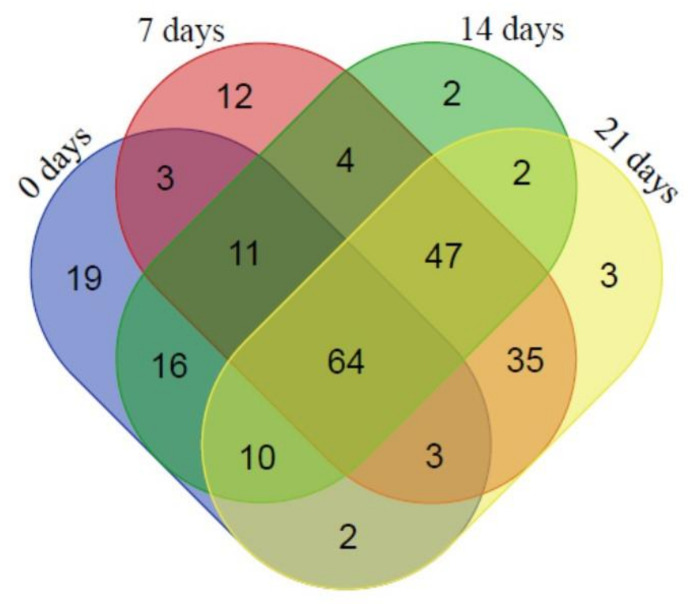
Venn-diagram of the shared and unique proteins present in mutant BIG8-1 plants before (Day 0) and after exposure to water deficit stress (Days 7, 14, 21).

**Figure 8 ijms-22-05314-f008:**
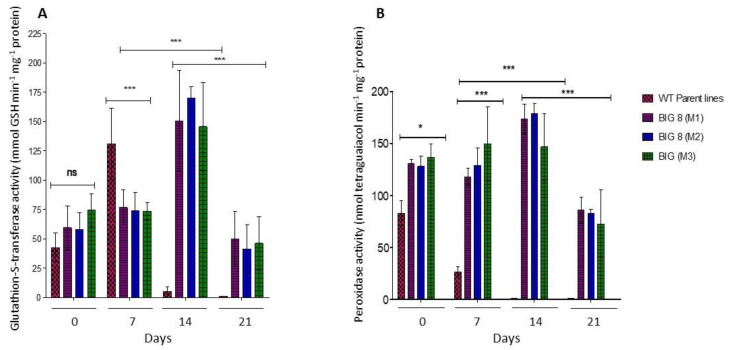
Changes in Glutathione-S-transferase (**A**) and Peroxidase (**B**) activities extracted from leaves of WT BIG8 and mutant BIG8-1 plants under well-watered (Day 0) and after exposure to water deficit stress (Day 7, 14, 21 PWS). POX activity was measured spectroscopically by the formation of tetraguaiacol monitored at 470 nm, while GST activity represents the formation of GS-DNB conjugate at 340 nm. Results shown represent the mean ± SD of three replications. Asterisks indicate statistically significant differences between WT BIG8 and mutant BIG8-1 plants * *p* ≤ 0.05. *** *p* ≤ 0.01 and ns refers to non-significance.

**Table 1 ijms-22-05314-t001:** Isolated leaf free amino acids from leave material of the BIG8 (WT) and BIG8-1 (Mutant) lines under well-watered (Day 0) and after induction of water deficit stress (Days 7, 14 and 21).

Genotype	Treatment	Free Amino Acid (FAA) Level															
		His	Ser	Arg	Gly	Asp	Glu	Thr	Ala	Pro	Lys	Tyr	Met	Val	Ile	Leu	Phe
**BIG8** **(WT) ***	**Day 0**	0.10	0.33	0.32	0.31	0.55	0.59	0.24	0.39	0.29	0.39	0.21	0.54	0.31	0.18	0.42	0.34
	**Day 7**	n.d	0.20	n.d	0.27	n.d	0.56	0.31	0.42	0.88	0.50	0.14	2.54	0.14	0.33	0.59	1.65
**BIG8-1** **(Mutant)**	**Day 0**	0.16	0.35	0.45	0.34	0.84	0.69	0.32	0.37	0.33	0.50	0.24	0.43	0.36	0.19	0.80	0.59
	**Day 7**	0.10	0.14	n.d	0.12	0.24	0.33	0.06	0.13	0.31	0.18	0.08	0.75	0.08	0.06	n.d	0.42
	**Day 14**	0.14	0.26	0.27	0.30	0.80	0.68	0.22	0.36	0.88	0.34	0.17	0.54	0.36	0.20	0.66	0.41
	**Day 21**	0.31	0.41	0.59	0.57	1.71	2.37	0.32	0.51	4.60	0.38	0.39	0.46	0.77	0.43	0.76	0.84

* Measurements could be only taken for the WT until Day 7, as it suffered irreversible damage and was dead by Day 9 (*n* = 3).

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
