# Peer review of "EMS Derived Wheat Mutant BIG8-1 (Triticum aestivum L.)—A New Drought Tolerant Mutant Wheat Line"

_ijms, 2021, doi:10.3390/ijms22105314_

Round 1

Reviewer 1 Report

None

Author Response

Response to Reviewer 1 Comments

Point 1: Comments and Suggestions for Authors “’None”’

Response 1: Since there are no comments or suggestions from Reviewer 1 to address, we just want to express our gratitude for taking the time to review our manuscript.

Reviewer 2 Report

Although an extensive and interesting work has been performed in the identification of a wheat mutant tolerant to drought, the work cannot be accepted in the present form due to different drawbacks. More importantly, further analysis of WT plants is required.

GENERAL

- Experimental model:

     - In WT plants, some parameters are measured until 7 days after induction of water deficit stress (see fig. 4B), whereas others are determined until 14 (see fig. 3A, 3B) or 21 days (see fig. 4A, fig. 8A, 8B). Once established, the different parameters should be analyzed at the same time points. Unless there is a justified cause (which must be indicated), all the parameters should be analyzed during the same time period.

      - The proteome analysis is only performed in mutant plants. WT plant should also be analyzed in order to draw conclusions about the modifications induced by mutation.

- Statistical analysis is sometimes missed. Furthermore, the way in which statistics is sometimes presented is not adequate (fig. 4A, fig. 8). Comparisons are not only necessary between WT and mutant plants. The evolution after induction of water deficit stress is also important.

- Some sentences are not precise or clear and should be rewritten (See below).

INTRODUCTION

- Lines 82-83: Revise reference format.

- Different inaccuracies are included in the last paragraph of page 2 and this information should be thoroughly reviewed.

     - Lines 86-87: “Photosynthesis consists of photosystem I (PSII), the cytochrome b6f complex and PSI” This is not correct, photosynthesis is much more.

     - Line 91: “… fluorescence, which is measurable as Fv/Fm”. Fluorescence is not measured as Fv/Fm.

     - Line 91-92: “Fv/Fm, which is the excitation energy by the PSII” Fv/Fm is the maximum quantum yield of PSII.

     - Line 97: The RuBisCo is an enzyme; therefore, it is evident that it catalyzes a reaction.

- Line 105-106: “However, stomatal closure is regarded as a necessity, as it limits internal water loss by increasing photorespiration…”. Photorespiration does not limit water loss.

MATERIAL AND METHODS

- Line 181, Why “physiological” assessment? In this sentence authors are only measuring leaf and shoot dimensions.

- Lines 183-186: Information about sampling is not clear. Please, clarify.

- Line 196: Authors do not measure photosynthesis; therefore, it cannot be included in the title of the 2.3. section.

- Lines 237-238: In Coomassie R-250 staining, proteins are usually dark blue and the background is clear. Do authors use a modified protocol? Please, clarify.

- Line 240-241:  Authors comment about contrast and brightness adjustment. These modifications are only allowed if they are applied to the complete gel. This information should be clarified and clearly indicated.

- Line 247: The technique used for amino acid separation should be indicated.

- Section 2.7: Plants used in the proteome analysis should be clearly indicated in this section. It is fundamental to specify the experimental model

- Line 253: The extraction protocol should be more clearly indicated. Author indicate that they use a modified method and three references are cited.

- Line 306: “All data were collected using three biological repeats (n = 3)…” Do authors use three different plants or three different leaves of a plant? Please, clarify

- Line 310: Why two post hoc test are used? Please, clarify.

RESULTS

- Why soil moisture content and RWC data are ±?

- Line 360: Authors determine chlorophyll concentration, no chlorophyll production. Please, be precise.

- Line 367: “A consistent and steady decline in chlorophyll fluorescence (Fv/Fm)”. This is not correct, as Fv/Fm is not chlorophyll fluorescence.

- Line 404: Why these sizes ranges? Please clarify

- Line 422: Fig. S2 is cited here, but Fig. 1S is missed.

- Line 461: It is Table 1, not Table 5.1.

DISCUSSION

- Line 508: “having more leaves, which was also broader — an attribute that is associated with drought adaption, for enhancement of photosynthesis, internal water retainment and gas exchange”. In adaptation to drought stress, shoot/root ratio tends to decrease (Taiz et al. 2015).

- Line 512: Please, revise the figure number.

- Line 514-515: “The visibly larger root mass creates a larger absorptive surface area per gram dry weight of root 96,97.” Larger root mass does not necessarily imply a larger absorptive surface per gram dry. Please, clarify.

- Lines 534-536: “It has proposed that the slow wilting phenotype can be attributed to the ability to conserve soil moisture under well-watered conditions100” How does plant contribute to conserve soil moisture? Please, clarify

- Line 540: “Chlorophyll a/b content was …”. According to material and methods and results, authors measured total chlorophyll concentration. It is important not to confuse with the chlorophyll a/b ratio.

- Line 555-556: “Hence, based on the chlorophyll content measured in the mutant plants relative to WT plants, a high rate of PSII efficiency (Fv/Fm) was observed (Figure 4)”. This sentence is confusing, not always low Fv/Fm values are linked to low chlorophyll contents.

- Line 568-569: “membranes and other cellular components resulting in eventual cell death (senescence after wilting)”. This sentence is confusing. Why (senescence after wilting) after cell death?

- Line 576: “Where POX catalyzes hydrogen for the oxidation of a wide range of substrates…”. POX uses hydrogen peroxide as substrate, not hydrogen.

- Line 611: “The SUMO attachment to substrates is a chronological enzyme reaction …”. Chronological? Please, clarify.

- Line 634-636: “Our results provide evidence that such drought-dependent oxidative processes were greatly suppressed in mutant plants”. Results shown in figure 8 do not support this statement. Please, revise.

- Lines 660-662: “The mutant plants remained viable for up to 21 days PWS when compared with WT plants that died after 10 days of water deficit stress.” However, in lines 334-336 authors write “… the WT plants visibly wilted after four days and had senesced leaves after seven days (Day 7, Figure 1B). The plant was severely senesced after nine days and completely dead after two weeks”. Please, clarify.

- Lines 662-664: “Reasons for better drought tolerance include that the mutant plants maintained leaf and root RWC at a much higher level for a longer period when compared to WT plants, despite low soil moisture availability. This is very likely due to broader leaves and denser root mass…” Broader leaves is not a phenotype associated to drought tolerance.

- Lines 662-665: “Reasons for better drought tolerance include that the mutant plants …. but also proteome reprogramming since unique proteins were expressed within BIG8-1, which likely contributed to altered metabolism.” This statement cannot be reached without analyzing the proteome of WT plants. Conclusions of the proteome analysis should be revised after proteome analysis of WT plants.

- When discussing the results obtained, authors should take into consideration that mutant plants maintain high RWC values longer tan WT plants. Therefore, they perceive the stress later.

FIGURES AND TABLES

- Fig. 1: Scale bars are missing.

- Fig. 2: X axes are not correct.

- Fig. 3: X axes should be Variable (units), i.e., “Time (days)”. In Y axes, it should be RWC, not RMC. Figure legend should be revised as it does not correspond to the represented data. Statistical analysis is missed.

- Fig. 4: X axes should be “Time (days)”.

Figure legend: In line 385 authors write “Each bar represents the mean ± SD of three replications”. It is not correct as each bar represents SD, not the mean.

- Fig. 4B: Why, differently to the other parameters, do authors represent stomata conductance in a scatter plot? Stomata conductance data should be represented as the mean and standard deviation in a line plot. Statistical analysis is missed. Therefore, no asterisks are included (see figure legend (line 390)). Units of stomata conductance are not correct. Figure legend: As previously indicated, Fv/Fm is not chlorophyll fluorescence, but the maximum quantum yield of PSII. Please, be precise.

- Fig. 5: The different gels should be clearly specified.

- Table 1: Units should be indicated. “Free amino acid (FAA) level” is not correct. Statistical analysis is missed.

- Supplementary figure: Figure legend is missed.

- Supplementary table 1: Data should be presented as the mean ± SD.

REFERENCES

The number of references is excessive. Many of them are not essential.

References

Taiz L, Zeiger E, Moller IM, Murphy A (2015) Plant Physiology and Development. 6th Ed. Sinauer Associates, Inc. Publishers.

Author Response

Response to Reviewer 2 Comments

Point 1: Although an extensive and interesting work has been performed in the identification of a wheat mutant tolerant to drought, the work cannot be accepted in the present form due to different drawbacks. More importantly, further analysis of WT plants is required.

Response 1: To address this comment/concern additional data was added as supplementary material for the purpose of this review which include the following:

(i) Supplementary table 2 with raw counts from the wildtype wheat (TugelaDN) and Mutant plants BIG8 and RYNO3936. The latter mutants were induced using different mutagens, namely EMS for BIG8 and Sodium Azide in the case of RYNO3936.

Table S5: Top list of identified proteins from WT Tugela DN, BIG8-1 and RYNO3936 before induction of water stress after analysis using the Scaffold Viewer 4 proteomics software (http://www.proteomesoftware.com/products/scaffold/; Searle, 2010). Indicated are the identified protein, accession number, molecular weight, protein grouping ambiguity, and level of significance after conducting an ANOVA (p < 0.00097) (n = 3).

(ii) Venn diagram of the comparison between the significantly expressed proteins in each of the wheat varieties (0h) (P<0.05).

Figure S3 Venn-diagram of the shared and unique proteins present in wildtype TugelaDN, and mutants BIG8-1 and RYNO3936 plants before (Day 0) exposure to water deficit stress.

(iii) The Table below show all the shared proteins between the mutants, but they share none with the wildtype wheat at 0h before induction of water deficit stress. (not included).

Table: Identities of the proteins shared between the mutant wheat lines BIG8 and RYNO3936.

Accession Number

Identified Proteins

W5G736_WHEAT

Uncharacterized protein OS=Triticum aestivum PE=4 SV=1

M8AVR4_AEGTA (+2)

20 kDa chaperonin, chloroplastic OS=Aegilops tauschii GN=F775_32594 PE=3 SV=1

M8AUX6_TRIUA

ATP synthase subunit alpha OS=Triticum urartu GN=atpA PE=3 SV=1           

C3VN75_WHEAT (+3)

Low molecular weight glutenin OS=Triticum aestivum GN=Glu-A3 PE=4 SV=1       

W5HAX6_WHEAT

Uncharacterized protein OS=Triticum aestivum PE=4 SV=1  

M7Z6F5_TRIUA

Glycine cleavage system H protein, mitochondrial OS=Triticum urartu GN=TRIUR3_12946 PE=3 SV=1           

O24400_WHEAT

Superoxide dismutase [Cu-Zn] OS=Triticum aestivum GN=SOD1.2 PE=2 SV=1      

W5F6W2_WHEAT

Uncharacterized protein OS=Triticum aestivum PE=4 SV=1  

M8BVC8_AEGTA

Cell division protease ftsH-like protein, chloroplastic OS=Aegilops tauschii GN=F775_28819 PE=3 SV=1           

W5GFX3_WHEAT (+1)

Uncharacterized protein OS=Triticum aestivum PE=4 SV=1  

N1R5T6_AEGTA

ATP synthase delta chain, chloroplastic OS=Aegilops tauschii GN=F775_06392 PE=3 SV=1         

M8BN30_AEGTA (+1)

30S ribosomal protein 2, chloroplastic OS=Aegilops tauschii GN=F775_28246 PE=4 SV=1         

M8AW52_AEGTA

Quinone oxidoreductase-like protein OS=Aegilops tauschii GN=F775_07275 PE=4 SV=1         

M8BNU5_AEGTA

Uncharacterized protein OS=Aegilops tauschii GN=F775_29168 PE=4 SV=1           

W5HCY0_WHEAT

Uncharacterized protein OS=Triticum aestivum PE=4 SV=1  

A0A077S298_WHEAT (+1)

Uncharacterized protein OS=Triticum aestivum GN=TRAES_3BF028000060CFD_c1 PE=3 SV=1  

Q41518_WHEAT

Single-stranded nucleic acid binding protein OS=Triticum aestivum GN=whGRP-1 PE=2 SV=1

Q0Q5D4_WHEAT (+1)

Globulin 1 OS=Triticum aestivum PE=4 SV=1        

In response to the reviewer’s comment/concern:  further analysis in this case is pointless as none of the significantly expressed proteins (P<0.05) is shared between the wildtype and the mutant lines.

GENERAL

Point 2: - Experimental model:

     - In WT plants, some parameters are measured until 7 days after induction of water deficit stress (see fig. 4B), whereas others are determined until 14 (see fig. 3A, 3B) or 21 days (see fig. 4A, fig. 8A, 8B). Once established, the different parameters should be analyzed at the same time points. Unless there is a justified cause (which must be indicated), all the parameters should be analyzed during the same time period.

Response 2: The reviewer has a valid point.  However, the data on the graphs showed that the WT plants were completely sensed /dead at day 7 post induction of water deficit stress. If this remains a concern, then those data point must be removed. In the case of Fv/Fm, no measures were detected on the device hence why it was not annotated on the graph.

Point 3:   - The proteome analysis is only performed in mutant plants. WT plant should also be analyzed in order to draw conclusions about the modifications induced by mutation.

Response 3: To address this comment, proteome data and a Venn diagram were included as supplementary data.  Also refer to the discussion under point 1.

Point 4:- Statistical analysis is sometimes missed. Furthermore, the way in which statistics is sometimes presented is not adequate (fig. 4A, fig. 8). Comparisons are not only necessary between WT and mutant plants. The evolution after induction of water deficit stress is also important.

Response 4: To address this comment, we made changes to these figures in that bars were added that shows the statical significance as per explanation in the Figure legend.

- Some sentences are not precise or clear and should be rewritten (See below).

INTRODUCTION

Point 5: - Lines 82-83: Revise reference format.

Response 5: The format has been revised as suggested.

Point 6: - Different inaccuracies are included in the last paragraph of page 2 and this information should be thoroughly reviewed.

Response 6: All the relevant inaccuracies as pointed out by the reviewer has been addressed in text.

Point 7:      - Lines 86-87: “Photosynthesis consists of photosystem I (PSII), the cytochrome b6f complex and PSI” This is not correct, photosynthesis is much more.

Response 7: To address this comment, the paragraph was changed to read as follow: “’When a plant experiences drought stress, the major components of the photosynthetic apparatus (photosystem II (PSII), the cytochrome b6f complex and PSI) become defec-tive 40, which is perpetuated by chlorophyll imbalance. Chlorophyll is essential for capturing light energy, which can transfer it for the use in photochemistry or releasing the remaining energy in the form of electromagnetic radiation, often referred to as fluorescence, which is an integral for the measurement of Fv/Fm 41. The latter is, therefore, the maximum quantum yield of PSII and an indication of the function of other thylakoid membrane protein complexes, collectively forming a photosynthetic network 42. Any compromise in photochemistry that goes beyond PSII is a causative effect of the downregulation of the photosynthetic process while stressed”’

Point 8:      - Line 91: “… fluorescence, which is measurable as Fv/Fm”. Fluorescence is not measured as Fv/Fm.

Response 8: Chlorophyll is essential for capturing light energy, which can transfer it for the use in photochemistry or releasing the remaining energy in the form of electromagnetic radiation often referred to as fluorescence, which is an integral for the measurement of Fv/Fm which is measurable as Fv/Fm.

https://www.researchgate.net/publication/340441653_Special_issue_in_honour_of_Prof_Reto_J_Strasser_-_Comparative_analysis_of_drought_stress_response_of_maize_genotypes_using_chlorophyll_fluorescence_measurements_and_leaf_relative_water_content.  Also see the reference to Zurek et al. (2014) and Kalaji et al. (2017) that states:” Chl fluorescence (ChlF) is electromagnetic radiation emitted by the Chl in plants. ChlF analysis is used to calculate the initial quantum yield of PSII QA reduction (Fv/Fm) and to quantify the performance of the electron transport chain as a performance index(PI), describing the ability of the photosynthetic apparatus to collect light energy and use it for photosynthetic electron transport. Fv/Fm has been widely used to describe the efficiency of PSII.”’

 Point 9:     - Line 91-92: “Fv/Fm, which is the excitation energy by the PSII” Fv/Fm is the maximum quantum yield of PSII.

Response 9:  To address this comment: “which is measurable as Fv/Fm 41”’ the latter is therefore the maximum quantum yield of PSII and an indication of the function of other thylakoid membrane protein complexes, collectively forming a photosynthetic network

Point 10:    - Line 97: The RuBisCo is an enzyme; therefore, it is evident that it catalyzes a reaction.

Response 10:  To address this comment, the paragraph was changed to read as follows: “’Ribulose-1,5-bisphosphate carboxylase/oxygenase (RuBisCO), a key enzyme in photo-synthesis, is further highly susceptible to water deficit stress. The RuBisCO protein as-similates carbon dioxide by catalysis, which results in the conversion of inorganic car-bon into organic compounds. Its relative abundance is important since the enzyme is a very slow-working enzyme. Large amounts of the enzyme are, therefore, required to achieve adequate photosynthesis and a direct relationship between the amount of Ru-BisCO and the rate of photosynthesis in higher plants, such as wheat, exists 47,48.”’

Point 11: - Line 105-106: “However, stomatal closure is regarded as a necessity, as it limits internal water loss by increasing photorespiration…”. Photorespiration does not limit water loss.

Response 11: To address this comment, the paragraph was rephrased to read as follows: “”A further inevitable reaction is the closure of stomata during drought stress. This becomes synonymous with a decline of CO2 uptake, limiting the carboxylation process 49. However, stomatal closure is regarded as a necessity, to further limit internal water loss50. The photosynthetic carbon fixation process is further compromised by capturing more light than can be actively processed. This process prompts a surge in the produc-tion of reactive oxygen species (ROS), which disrupts photosynthesis51’”

MATERIAL AND METHODS

Point 12: - Line 181, Why “physiological” assessment? In this sentence authors are only measuring leaf and shoot dimensions.

Response 12: To address this comment the sentence was rephrased to read: “”The physiological assessment of leaf (length, width) and shoot (height) dimensions was determined with a measuring tape (unit of measurement in cm).”

Point 13: - Lines 183-186: Information about sampling is not clear. Please, clarify.

Response 2: The authors disagree as the sampling is described in detail and read as follow: “”The leaf (length, width) and shoot (height) dimensions were determined with a measuring tape (unit of measurement in cm). Plant height was measured from the tip of the tiller to the ground (n = 20).  Triplicate samples were prepared for the measurement of relative water content (RWC). Leaf and root material were assessed at each of the respective time points (Days 0, 7, 14, 21) using three similar-sized leaves and six replicates. The third leaf was cut to determine the fresh weight (FW), and then placed in deionized water for 24 h at room temperature under low-light conditions. After soaking, leaves were blotted dry with tissue paper, and turgidity measured (TW). The same leaf was subjected to complete dehydration by placing it in a benchtop oven at 80 °C for 16 h to quantify the dry weight (DW). RWC was calculated using the following equation: RWC (%) = (FW-DW) / (TW-DW) × 100%68,69. Soil samples (n = 3) were collected at a depth of 150 mm and dried in an oven at 105°C for 48 h after which the soil was again weighed and the gravimetric soil moisture content determined 70. Wilting of both the mutant and the WT plants was assessed at each of the time points following the categories described71”’

Point 14: - Line 196: Authors do not measure photosynthesis; therefore, it cannot be included in the title of the 2.3. section.

Response 14: To address this comment, the heading was changed to “’ Chlorophyll fluorescence (Fv/Fm)stomatal conductance, and chlorophyll content”’

Point 15: - Lines 237-238: In Coomassie R-250 staining, proteins are usually dark blue and the background is clear. Do authors use a modified protocol? Please, clarify.

Response 15: To explain, this is not a modified protocol and not ordinary SDS_PAGE gels but ZYMO gels which is blue – i.e., the substrate and the protein, when present degrade the substrate to give translucent bands.

Point 16: - Line 240-241:  Authors comment about contrast and brightness adjustment. These modifications are only allowed if they are applied to the complete gel. This information should be clarified and clearly indicated.

Response 16: To address this comment the following explanation is provided: “Gels were digitalised using Gel Doc XR+ System and imported to Microsoft PowerPoint 2016 (KB4011564) 64-Bit Edition, where the entire gel image was adjusted to +40% and brightness to +20% and finally cropped for presentation purposes

Point 17:  Line 247: The technique used for amino acid separation should be indicated.

Response 17:  The technique is comprehensively/in detail described in the referenced article. To add the detailed description will add additional information to the material and methods section. This is contradictory to the comments of Reviewer 3, who advised reducing the detailed description of this section. To quote: ‘The methodology was thorough and comprehensive, it could have been a bit more concise. The article almost came across as a discussion on methodology, rather than discussing the virtues of BIG8-1. This is what I was expecting from the title of the article.”.  

Point 18: - Section 2.7: Plants used in the proteome analysis should be clearly indicated in this section. It is fundamental to specify the experimental model

Response 18: Similarly, as in point 17, the description is sufficient and additional information will further be in contradiction to Reviewer 3’s comment.  As is, it reads:  “’All proteome analyses were conducted using three biological repeats (n=3). Plants were exposed to well-watered (Day 0) and water deficit stress conditions (Days 7, 14, 21) after which leaf protein (n=3) was extracted using a modified method 37. After extraction, the pellet was lyophilized for two hours and stored at -80°C until further use”’

Point 19: - Line 253: The extraction protocol should be more clearly indicated. Author indicate that they use a modified method and three references are cited.

Response 19: Similarly, as in point 17 and 18, the description is sufficient and additional information will further be in contradiction to Reviewer 3’s comment.  As is, it reads:  “’All proteome analyses were conducted using three biological repeats (n=3). Plants were exposed to well-watered (Day 0) and water deficit stress conditions (Days 7, 14, 21) after which leaf protein (n=3) was extracted using a modified method 37. After extraction, the pellet was lyophilized for two hours and stored at -80°C until further use”

Point 20: - Line 306: “All data were collected using three biological repeats (n = 3)…” Do authors use three different plants or three different leaves of a plant? Please, clarify

Response 20: To clarify n=3 always implies 3 independent biological repeats = THUS 3 independent/different plants. Three different leaves will be the SAME biological repeat. In all instances n=3 will imply three independent biological repeats and thus independent plants.

Point 21: - Line 310: Why two post hoc test are used? Please, clarify.

Response 21:

  • By definition: Tukey's test compares the means of every treatment to the means of every other treatment.
  • By definition: Dunnett's test is used when we want to compare one group (usually the control treatment) with the other groups.

The test was chosen based on the experiment question. When the control died out, especially for activity experiments, it was preferred to do a turkey’s test (Fig 8). However, experiments measuring non-activity facets such as physiological change, we did Dunnets test to understand how the plant change over time and not how it compares to the control ( Day 0 to day 21).

RESULTS

Point 22: - Why soil moisture content and RWC data are ±?

Response 22: The reason for use approximation was used, due to the small variable of RWC and gravermetric readings when conducting such an experiment. Generally, when repeating the weight measurements of RWC or gravimetrics, it will differ in a very small range in each repeat hence the use of approximation.

Point 23: - Line 360: Authors determine chlorophyll concentration, no chlorophyll production. Please, be precise.

Response 23: “”production” was changed to “concentration” and now reads: “”Next chloropyll concentration was investigated in the two types of plants’”

Point 24: - Line 367: “A consistent and steady decline in chlorophyll fluorescence (Fv/Fm)”. This is not correct, as Fv/Fm is not chlorophyll fluorescence.

Response 24: The sentence was changed to read: “’A consistent and steady decline in the maximum quantum yield of PSII (Fv/Fm)”

Point 25: - Line 404: Why these sizes ranges? Please clarify

Response 25: These are crucial to indicate the size of these proteases. These sizes allude to the fact that these COULD be other proteases forming part of the four major classes of proteases based on their catalytic reaction mechanisms: cysteine, serine, aspartate, and metalloproteases (refer to the MEROPS database). This was clarified in the manuscript.

Point 26: - Line 422: Fig. S2 is cited here, but Fig. 1S is missed.

Response 26: To address this comment, the reference to the figure was added and it now reads:” Supplementary Figures S1 and S2”

Point 27: - Line 461: It is Table 1, not Table 5.1.

Response 27: Line 637 The table reference was corrected and now reads “’Table 1”’

DISCUSSION

Point 27: - Line 508: “having more leaves, which was also broader — an attribute that is associated with drought adaption, for enhancement of photosynthesis, internal water retainment and gas exchange”. In adaptation to drought stress, shoot/root ratio tends to decrease (Taiz et al. 2015).

Response 27: in text references (93-95) was replaced by Taiz et al. 2015 as suggested.

Point 28: - Line 512: Please, revise the figure number

Response 28: The figure number was revised to 1D.

.

Point 29: - Line 514-515: “The visibly larger root mass creates a larger absorptive surface area per gram dry weight of root 96,97.” Larger root mass does not necessarily imply a larger absorptive surface per gram dry. Please, clarify.

Response 29:  To clarify: the cited references emphasize and support this statement – but it was phrase with caution: ”The visibly larger root mass presumably creates a larger absorptive surface area per gram dry weight of root 96,97.” However, further investigation is needed in this regard”’

Point 30: - Lines 534-536: “It has proposed that the slow wilting phenotype can be attributed to the ability to conserve soil moisture under well-watered conditions100” How does plant contribute to conserve soil moisture? Please, clarify

Response 30: To address this comment, the following changes were made and now reads: “’The slow wilting phenotype can be attributed to the ability to conserve water from the soil under well-watered conditions100

Point 31: - Line 540: “Chlorophyll a/b content was …”. According to material and methods and results, authors measured total chlorophyll concentration. It is important not to confuse with the chlorophyll a/b ratio.

Response 31: In complete agreement - it was important to mentioned Chlorophyll a/b content as this is use when calculating total Chlorophyll.

Point 32: - Line 555-556: “Hence, based on the chlorophyll content measured in the mutant plants relative to WT plants, a high rate of PSII efficiency (Fv/Fm) was observed (Figure 4)”. This sentence is confusing, not always low Fv/Fm values are linked to low chlorophyll contents.

Response 32: To address this comment the following changes were made: ” Photosynthetic pigments are instrumental in allowing the plant to absorb energy from sunlight and foliar chlorophyll content is a determining factor in photosynthetic rates 107. Although we did not demonstrate a statical relationship between chlorophyll content and the rate of PSII efficiency (Fv/Fm), our data does show a concomitant change of these two physiological parameters over time. Therefore, based on the chlorophyll content measured in the mutant plants relative to WT plants, a high rate of PSII efficiency (Fv/Fm) was observed (Figure 4).”’

Point 33: - Line 568-569: “membranes and other cellular components resulting in eventual cell death (senescence after wilting)”. This sentence is confusing. Why (senescence after wilting) after cell death?

Response 33: To address this comment the phrase “(senescence after wilting)” was  removed to avoid further confusion.

Point 34: - Line 576: “Where POX catalyzes hydrogen for the oxidation of a wide range of substrates…”. POX uses hydrogen peroxide as substrate, not hydrogen.

Response 34: “hydrogen”’ was replaced with “hydrogen peroxide”’

Point 35: - Line 611: “The SUMO attachment to substrates is a chronological enzyme reaction …”. Chronological? Please, clarify.

Response 35: “chronological”” was replaced with “sequential”’

Point 36: - Line 634-636: “Our results provide evidence that such drought-dependent oxidative processes were greatly suppressed in mutant plants”. Results shown in figure 8 do not support this statement. Please, revise.

Response 36: To address the comment, we removed the words “greatly suppressed” and the statement has therefore been changed and now reads as follows: “Our results provide evidence that such drought-dependent oxidative processes were  likely reduced in the mutant plants”

Point 37: - Lines 660-662: “The mutant plants remained viable for up to 21 days PWS when compared with WT plants that died after 10 days of water deficit stress.” However, in lines 334-336 authors write “… the WT plants visibly wilted after four days and had senesced leaves after seven days (Day 7, Figure 1B). The plant was severely senesced after nine days and completely dead after two weeks”. Please, clarify.

Response 37: To address this comment, the correction has been made on the figure 1, and in line 334-336.

Point 38: - Lines 662-664: “Reasons for better drought tolerance include that the mutant plants maintained leaf and root RWC at a much higher level for a longer period when compared to WT plants, despite low soil moisture availability. This is very likely due to broader leaves and denser root mass…” Broader leaves is not a phenotype associated to drought tolerance.

Response 38: To address this comment, the sentence was changed to read: “This is very likely due to the denser root mass but also proteome reprogramming since unique proteins were expressed within BIG8-1, which likely contributed to altered metabolism”

Point 39: - Lines 662-665: “Reasons for better drought tolerance include that the mutant plants …. but also proteome reprogramming since unique proteins were expressed within BIG8-1, which likely contributed to altered metabolism.” This statement cannot be reached without analyzing the proteome of WT plants. Conclusions of the proteome analysis should be revised after proteome analysis of WT plants.

Response 39: Supplementary data was added in support of this statement – please refer to Supplementary Table S2 and Figure S3.

Point 40: - When discussing the results obtained, authors should take into consideration that mutant plants maintain high RWC values longer tan WT plants. Therefore, they perceive the stress later.

Response 40: This is likely that these plants experience stress later, however this was not an objective that was set.

FIGURES AND TABLES

Point 41: - Fig. 1: Scale bars are missing.

Response 41: Image has been updated as requested by the reviewer.

Point 42: - Fig. 2: X axes are not correct.

Response 42: X axes heading now shows “Time (days)” as requested.

Point 43: - Fig. 3: X axes should be Variable (units), i.e., “Time (days)”. In Y axes, it should be RWC, not RMC. Figure legend should be revised as it does not correspond to the represented data. Statistical analysis is missed.

Response 43: X axes heading now shows “Time (days)” as requested. Requested changes had been made.

Point 44: - Fig. 4: X axes should be “Time (days)”.

Response 44:  X axes heading now shows “Time (days)” as requested.

Point 45: Figure legend: In line 385 authors write “Each bar represents the mean ± SD of three replications”. It is not correct as each bar represents SD, not the mean.

Response 45: The figure legend was changed as suggested.

Point 46: - Fig. 4B: Why, differently to the other parameters, do authors represent stomata conductance in a scatter plot? Stomata conductance data should be represented as the mean and standard deviation in a line plot. Statistical analysis is missed. Therefore, no asterisks are included (see figure legend (line 390)). Units of stomata conductance are not correct. Figure legend: As previously indicated, Fv/Fm is not chlorophyll fluorescence, but the maximum quantum yield of PSII. Please, be precise.

Response 46: Why, differently to the other parameters, do authors represent stomata conductance in a scatter plot? YES as mentioned in the description. Stomata conductance data should be represented as the mean and standard deviation in a line plot. Stomata conductance is presented as the scatter plot with its linear regression – however perhaps we should mentioned that dotted line refers to the linear regression?

 Statistical analysis is missed. The linear regression is the analysis here - by definition in the context of our study the regression line models the relationship between time(days) which is the “scalar response” and measured values which is the Stomata conductance (“explanatory variables”).  Therefore, no asterisks are included (see figure legend (line 390)) = has been corrected. Units of stomata conductance are not correct = has been corrected. . Figure legend: As previously indicated, Fv/Fm is not chlorophyll fluorescence, but the maximum quantum yield of PSII = has been corrected. Please, be precise.

Point 47: - Fig. 5: The different gels should be clearly specified.

Response 47: This has been corrected = labelling each A, B and C

Point 48: - Table 1: Units should be indicated. “Free amino acid (FAA) level” is not correct. Statistical analysis is missed.

Response 48: These are absolute values and based on two biological repeats that was conducted over time. No technical repeats were done.

Point 49: - Supplementary figure: Figure legend is missed.

Response 49: Figure S2. Cluster image generated by Java TreeView (Saldanha, 2004) of the protein obtained after LC-ESI-MS/MS analysis of total protein isolated from BIG8-1 before (Day 0), and after induction of water stress (Days 7 and 14), and after recovery and regrowth (Day 21). Red bands show up-regulated proteins, whereas green bands show down-regulated proteins.

Point 50: - Supplementary table 1: Data should be presented as the mean ± SD.

Response 50: The requested calculations were added to Supplementary Table S2

REFERENCES

Point 51: The number of references is excessive. Many of them are not essential.

Response 51: Several references were removed from the list as suggested.

(80) Damerval, C.; Vienne, D. D.; Zivy, M.; Thiellement, H. Technical Improvements in Two-Dimensional Electrophoresis Increase the Level of Genetic Variation Detected in Wheat-Seedling Proteins. Electrophoresis 1986, 7 (1), 52–54. https://doi.org/10.1002/elps.1150070108.

(81)     Wang, W.; Vignani, R.; Scali, M.; Cresti, M. A Universal and Rapid Protocol for Protein Extraction from Recalcitrant Plant Tissues for Proteomic Analysis. Electrophoresis 2006, 27 (13), 2782–2786. https://doi.org/10.1002/elps.200500722.

(82)     Wang, X.; Li, X.; Deng, X.; Han, H.; Shi, W.; Li, Y. A Protein Extraction Method Compatible with Proteomic Analysis for the Euhalophyte Salicornia Europaea. Electrophoresis 2007, 28 (21), 3976–3987. https://doi.org/10.1002/elps.200600805.

(93) Nezhadahmadi, A.; Prodhan, Z. H.; Faruq, G. Drought Tolerance in Wheat. The Scientific World Journal 2013, 1, 12–21. https://doi.org/10.1155/2013/610721.

(94)     Peña-Rojas, K.; Aranda, X.; Joffre, R.; Fleck, I. Leaf Morphology, Photochemistry and Water Status Changes in Resprouting Quercus Ilex during Drought. Functional Plant Biol. 2005, 32 (2), 117–130. https://doi.org/10.1071/FP04137.

(95)     Shao, H.-B.; Chu, L.-Y.; Jaleel, C. A.; Zhao, C.-X. Water-Deficit Stress-Induced Anatomical Changes in Higher Plants. Comptes Rendus Biologies 2008, 331 (3), 215–225. https://doi.org/10.1016/j.crvi.2008.01.002.

Reviewer 3 Report

  1. The paper was informative, however, some minor grammatical and syntax errors were detected.
  2. The methodology was thorough and comprehensive, it could have been a bit more concise. The article almost came across as a discussion on methodology, rather than discussing the virtues of BIG8-1. This is what I was expecting from the title of the article.
  3. It would also have been interesting to know how BIG8-1 grain quality and "potential yield" stacked up against other varieties and cultivars. The authors could have acknowledged this or at least clarified that this was still in progress (in light of the title and introduction with its emphasis on yield). Was this something that the Authors considered?      

Author Response

Response to Reviewer 3 Comments

Point 1: The paper was informative, however, some minor grammatical and syntax errors were detected.

Response 1: The manuscript was revised to address the “’grammatical and syntax error”’ issues.

Point 2: The methodology was thorough and comprehensive, it could have been a bit more concise. The article almost came across as a discussion on methodology, rather than discussing the virtues of BIG8-1. This is what I was expecting from the title of the article.

Response 2: Due to the fact that Reviewer 2 requested more information/detailed description, we didn’t reduce the methodology section nor were more information added.

Point 3: It would also have been interesting to know how BIG8-1 grain quality and "potential yield" stacked up against other varieties and cultivars. The authors could have acknowledged this or at least clarified that this was still in progress (in light of the title and introduction with its emphasis on yield). Was this something that the Authors considered

Response 3: We did measure sedimentation value of the grounded grain and didn’t find any significant differences, but yield is higher in the mutants than the WT. But since the results on the sedimentation value and yield are part of another study, it wasn’t included.

Lines

Uncorrected Sedimentation value(ml)

% Moisture content

Corrected Sedimentation value(ml)

WT

19,3

10,9

18,66

8.1

21

11

20,29

Round 2

Reviewer 2 Report

As indicated in the previous revision, the manuscript presents important drawbacks. Although some comments have been taken into consideration, very important drawbacks have not been solved. Therefore, in my opinion, this work cannot be accepted for publication.